# Industry views of the UK Soft Drinks Industry Levy: a thematic analysis of elite interviews with food and drink industry professionals, 2018–2020

Catrin P Jones [ID],[1] Hannah Forde,[1,2] Tarra L Penney,[3] Dolly van Tulleken,[1] Steven Cummins [ID],[4] Jean Adams [ID],[1] Cherry Law,[5] Harry Rutter [ID],[6] Richard Smith,[7] Martin White[1]

For numbered affiliations see end of article.

**Correspondence to**
Dr Catrin P Jones;
Catrin.Jones@mrc-epid.cam.ac.uk

## ABSTRACT

**Objectives** The UK Soft Drinks Industry Levy (SDIL), implemented in 2018, has been successful in reducing the sugar content and purchasing of soft drinks, with limited financial impact on industry. Understanding the views of food and drink industry professionals involved in reacting to the SDIL is important for policymaking. However, their perceptions of the challenges of implementation and strategic responses are unknown. The aim of this study, therefore, was to explore how senior food and drink industry professionals viewed the SDIL.

**Design** We undertook a qualitative descriptive study using elite interviews. Data were analysed using Braun and Clarke's thematic analysis, taking an inductive exploratory and descriptive approach not informed by prior theory or frameworks.

**Setting and participants** Interviews were conducted via telephone with 14 senior professionals working in the food and drink industry.

**Results** Five main themes were identified: *(1) a level playing field…for some*; industry accepted the SDIL as an attempt to create a level playing field but due to the exclusion of milk-based drinks, this was viewed as inadequate, *(2) complex to implement, but no lasting negative effects*; the SDIL was complex, expensive and time consuming to implement, with industry responses dependent on leadership buy-in, *(3) why us?—the SDIL unfairly targets the drinks industry*; soft drinks are an unfair target when other categories also contain high sugar, *(4) the consumer is king*; consumers were a key focus of the industry response to this policy and *(5) the future of the SDIL*; there appeared to be a wider ripple effect, which primed industry to prepare for future regulation in support of health and environmental sustainability.

**Conclusions** Insights from senior food and drink industry professionals illustrate how sugar-sweetened beverage taxes might be successfully implemented and improve understanding of industry responses to taxes and other food and drink policies.

**Trial registration number** ISRCTN18042742.

## INTRODUCTION

Diet-related non-communicable diseases are a major and growing problem, responsible for over 11 million deaths globally each year.[1] Sugar consumption is of particular concern, with the WHO recommending member states introduce sugar-sweetened beverage (SSB) taxes.[2] Reviews suggest that they reduce sales of, increase prices of and encourage reformulation of SSBs,[3–5] and over 100 SSB taxes have been implemented worldwide covering 52% of the world's population.[6] SSB taxes have a variety of designs with 87% excise taxes.[6] The WHO recommend that a tiered SSB tax be introduced in companies with high administrative capacity, similar to that which has been introduced in the UK.[2] The Soft Drinks Industry Levy (SDIL) was announced on 16 March 2016 and implemented in the UK on 6 April 2018. According to the budget speech by George Osborne, Chancellor of the Exchequer at the time, it was designed to incentivise manufacturers of SSBs to reformulate their products[7] via charging a levy on soft drinks produced by companies when they leave the warehouse or when imported into the country.[8] Integrated in August 2016 as part of the UK Government's Childhood Obesity: A

## STRENGTHS AND LIMITATIONS OF THIS STUDY

⇒ This qualitative study explored how senior food and drink industry professionals viewed the Soft Drinks Industry Levy.
⇒ We undertook elite interviews with 14 professionals working in the food and drink industry, who have often been difficult to recruit in other studies.
⇒ Braun and Clarke's thematic analysis, taking a descriptive approach, was used to analyse the data.
⇒ Elite interviewing methods allow for the building of relationships to elicit meaningful responses from participants.
⇒ A limitation of this work is that interviews were carried out over a long period of time due to challenges in recruitment.

> **Box 1  Soft Drinks Industry Levy particulars[7]**
>
> Eligible drinks:
> ⇒ ≥8 g total sugar per 100 mL charged at 24 pence per litre.
> ⇒ ≥5 g and <8 g total sugar per 100 mL charged at 18 pence per litre.
> Exemptions:
> ⇒ Drinks containing more than 75% milk or 1.2% alcohol.
> ⇒ Alcohol substitute drinks.
> ⇒ Powdered drinks.
> ⇒ 100% fruit juices.
> ⇒ Manufacturers selling under one million litres of drinks per year.[50]

Plan For Action,[9] the SDIL consists of two tiers (for particulars of the tax, see box 1). A public consultation on the proposals between August and October 2016 set out the plans for the tiers and exclusions as described in box 1. Few changes were made as a result of this consultation and the SDIL was given royal assent on 27 April 2017. The government published a second chapter of its childhood obesity plan in 2018, which suggested the SDIL may be extended to milk-based drinks, though this has not yet occurred.[10]

The SDIL was one of the first SSB fiscal interventions explicitly designed to incentivise reformulation.[7 11 12] This aim was largely achieved, substantially reducing overall SSB sugar content, and inducing a major shift of drinks from the higher levy tier to the lower tier and untaxed bracket between 2016 and 2018 (Scarborough *et al*, 2020). Reformulation is reflected in purchases of sugar from SSBs.[13] Prior to implementation of the SDIL, the food and drinks industry (hereafter referred to as 'industry') viewed the SDIL as having a potentially negative impact on profits resulting in job losses.[14–16] A negative stock market reaction to the SDIL announcement was observed, but this only lasted 2 days.[17] Similarly, a negative impact on company domestic turnover was observed following the announcement of the SDIL, but this was resolved by the time of its implementation.[18]

Critical to the success of the SDIL is the implementation of and reaction to the regulations by the drink industry. Therefore, it is important to understand the perspectives of the industry as well as those who work in it regarding the implementation of such taxes. Previous work has investigated industry perspectives of the SDIL expressed through the news media[14 19–22] and the views of industry, civil society and academic participants on how marketing changed in response to the SDIL.[23] A notable gap in the literature, however, is perspectives of the SDIL from the commercial sector, not communicated publicly through news media nor focused solely on marketing responses. Important learning can be obtained by exploring the perspectives of commercial actors involved in responding to regulation. Interviews with senior members of industry can help examine the impact of the SDIL on both the soft drinks industry and wider food and drink industry, an avenue not previously explored. This study therefore aimed to address these knowledge gaps and inform policymaking by exploring the perspectives of senior industry professionals regarding the UK SDIL.

## METHODS

### Study design

This study adopted a qualitative descriptive design involving elite interviews with senior industry professionals.

### Methodological orientation

This research took an experiential qualitative approach, within a critical realist position. Participant perspectives and perceptions were prized over researcher interpretations, and reality was derived from our participants' words and meaning, rather than a reality constructed through researchers' interpretation of their words.[24] A descriptive approach was used to explore how the SDIL was viewed from the position of our participants.

### Research team

MW, SC, JA, RS and HR secured funding for the overall evaluation of the SDIL within which this study formed a part.[25] Interviews were conducted by postdoctoral research associates TLP and CPJ. TLP led the design of data collection and CPJ led the design of the analysis. MW and SC provided guidance on the design of both elements. TLP and CPJ recruited and interviewed participants. CPJ led the analysis with support from HF. HF conducted secondary coding to support theme generation and interpretation. All authors previously mentioned, as well as DT and CL were involved in data analysis and interpretation, as well as drafting this manuscript.

### Participants

Senior professionals from the soft drinks, food and other drinks industries were recruited to this study using purposive and snowball sampling. We adopted 'elite interviewing' methods to maximise involvement of senior professionals in positions of influence within their organisation and with high levels of responsibility.[26] This technique provides a series of strategies to support recruitment of difficult to access key participants.[27] The principles of elite interviewing were used to inform recruitment including stronger emphasis on the maintenance of trust, importance of interview tone of the interview, preparing appropriately, and engaging in and tailoring dialogue relevant to each informant, more so than in traditional interviews.[28 29]

Individuals were considered eligible to participate based on the following criteria: (a) currently or previously held a high-level industry position (at the managerial, director or chief officer level), (b) their organisation and their professional role were directly or indirectly impacted by the SDIL and (c) they could provide a novel perspective, determined by their job role or the company they work for not previously heard in our interviews. Recruitment typically involved an email introduction by a member of the

team or informant contact, although CPJ also attended industry food events and recruited face to face. Initial contact was followed by an informal telephone conversation with TLP or CPJ to discuss the research purpose, team and informant interests and perspectives, ultimately proceeding to full participation via telephone interview. Recruitment ceased when networks were exhausted, and no further contacts were identified.

## Data collection

Telephone interviews were conducted from June 2018 to June 2020. Participant information sheets were sent to potential participants prior to participating in the informal discussion. Informed consent was obtained verbally prior to commencement of the formal telephone interview, which was digitally audio recorded. Interviews were undertaken using a minimally structured topic guide containing three broad areas of inquiry: (a) can you tell me about your role and organisation?, (b) can you tell me about your sector as a whole? and (c) what do you know about the UK SDIL and its impacts? Elite interviewing necessitates informed and adaptive dialogue,[28 30] meaning participants could engage in ways most relevant to their specific expertise or experiences within these broad areas. Interviews were transcribed verbatim by a trusted external company, and transcripts were checked against the audio files by CPJ to identify any inaccuracies. Transcripts were anonymised prior to analysis by removing names of people, organisations and brands.

## Analysis

Analysis commenced once all interviews had been conducted and transcribed. Braun and Clarke's thematic analysis was used, taking an inductive exploratory and descriptive approach not informed by any prior theory or framework.[31] This approach is flexible due to lack of alignment with specific epistemological and ontological stances.[32 33] Six analytical steps were conducted: (1) familiarisation, (2) data coding, (3) initial theme generation, (4) theme development and review, (5) theme refining, defining and naming and (6) writing up.

CPJ listened to audio files and read transcripts at least two times to become familiar with them, while making notes on initial impressions and patterns (step 1). Following familiarisation, CPJ worked systematically through the entire data set and conducted complete coding of all data, in which segments of data were given a label to describe their area of interest. Coding was supported by NVivo software V.12. Semantic codes were derived directly from participants' speech or codes where phrases of speech were brief enough to be directly coded (step 2). CPJ then sorted these initial codes into concise categories (overarching codes), which clearly described the content of the data (step 3). A reflective diary was kept throughout the coding process by CPJ to note reflections on findings and to ensure a data-driven analytic process. Please see online supplemental file 1 for a detailed account of reflexivity.

HF also familiarised itself with the transcripts (step 1) and then examined CPJ's coding to ensure the codes were data-driven with as little interpretation as possible (step 2). CPJ then collated codes that shared a common pattern into themes (step 3). Again, CPJ and HF met to discuss and refine the themes to ensure they were descriptive with minimal interpretation (step 4).

A document containing themes, codes within them and extensive anonymised quotes was shared with all co-authors in two phases: phase 1 March 2022 and phase 2 October 2022 (step 5). This data clinic aimed to minimise researcher interpretation. A document presented theme descriptions and asked co-authors to answer the following questions for each theme: (1) is the theme descriptive?, (2) does the theme represent the data accurately? and (3) what do you think the theme tells us about the SDIL from the perspective of industry? JA, DT and CL completed the data clinic document in phase 1. Themes were amended based on their reflections and the document was updated in October 2022. SC, MW, HR and RS completed the data clinic form in phase 2. Final themes and the manuscript were written up by CPJ and reviewed by all co-authors (step 6).

## Patient and public involvement

This study is part of the 'Evaluation of the health impacts of the UK Treasury Soft Drinks Industry Levy (SDIL)' funded by NIHR (award no. 16/130/01). Project oversight is provided by an independent study steering committee (ISSC) which contains members of the public. The ISSC for the overall project met biannually from 2017 to 2023 and were asked to provide advice on methodology as well as interpretation of our findings.

## RESULTS

Fourteen participants were recruited (table 1). Participants' roles within organisations were diverse; chief officers, directors and managers with overall responsibility or with specialist responsibilities for finance, strategy, operations, marketing, public relations or nutrition. Interviews ranged in length from 26 to 62 min. Six additional

**Table 1** Participant details

| Sector category | n |
| --- | --- |
| Drink manufacturers | 4 |
| Food and drink manufacturers | 3 |
| Supermarkets | 3 |
| Industry associations | 1 |
| Out-of-home* food and drink manufacturers | 1 |
| Out-of-home retailers | 1 |
| Advertising consultants | 1 |

*The out-of-home sector is generally considered to be any outlet where food or drink is prepared in a way that means it is ready for immediate consumption, on or off the premises.[51]

---

> **Box 2   Theme and subtheme summary**
>
> Theme 1: a level playing field…for some
> ⇒ The SDIL created a level playing field.
> ⇒ Milk-based drinks increased the complexity in the out-of-home sector.
> ⇒ Challenges for supermarkets with large product portfolios.
> Theme 2: complex to implement but no lasting negative effects
> ⇒ Complexities in strategic response—price and product are key.
> ⇒ Leadership buy-in dictates strategic response.
> ⇒ Global companies and internal systems.
> ⇒ Contradictory government messaging.
> ⇒ Few long-lasting negative effects and the SDIL provided opportunities.
> Theme 3: why us?—the SDIL unfairly targets the drinks industry
> ⇒ Sugary drinks in isolation were unfair targets for regulation.
> ⇒ Distrust of government's motivations to introduce the SDIL.
> Theme 4: the consumer is king
> ⇒ Consumer response to product changes resulting from the SDIL.
> ⇒ Consumer momentum towards healthier products.
> Theme 5: the future of the SDIL
> ⇒ Extending to milk-based and fruit-based drinks.
> ⇒ Impact on the wider food and drink industry and on other sectors.
> ⇒ Proposal to reverse the SDIL.

participants were approached and took part in informal discussions; three did not participate due to scheduling issues, and three refused to take part. Five inductively derived, interlinked themes and 15 subthemes were identified (box 2).

### Theme 1: a level playing field…for some
#### The SDIL created a level playing field
Industry professionals accepted that the SDIL helped create a level playing field, where no organisation lost out by taking action on health that their competitors did not.

> … legislation level playing fields is so important and that's why with these big public health initiatives…I'm actually really quite pro government intervention.—Supermarket

Soft drinks manufacturers also discussed that the 2 years to prepare for the implementation of the SDIL was sufficient and they were happy they could develop an adequate response within that time.

> I'm not aware of any significant implementation or challenges that our members have encountered, I mean they did have time to adapt, the legislation was published in good time to allow them to understand exactly what they would be required to do.—Trade association

However, participants also stated that a lack of understanding and consultation from government meant a 'true' level playing field for all sectors involved in the sales of sugary drinks had not been not achieved.

> …you want to really do it smartly so everybody feels they're 100% equally affected and you don't get

this…'my product is in scope, your product is out of scope'…it doesn't create the sense of unilateral 'let's do this'…which is what it should be, if that makes sense.—Food and drink manufacturer

The lack of consultation by the government with sectors who were not soft drinks manufacturers (eg, out of home retailers) and the exclusion of milk-based sugary drinks led to this perception.

> …milk-based drinks often carried bigger serving sizes and had more total sugar in them than any of our products would. They were excluded from the levy as well which looked like a big shortcoming.—Drink manufacturer

#### Milk-based drinks increased the complexity in the out-of-home sector
Interviewees explained that, from their perspective, the government did not think clearly about the technical implications for retailers and out-of-home sector and that it was easier for soft drink manufacturers to respond to the levy than it was for other industry actors.

> …I don't think they understood the ways of working and the preparation methods in the out-of-home sector…—Out-of-home food and drink manufacturer

A high level of complexity within the out-of-home sector to manufacture and produce drinks for immediate consumption led to higher implementation costs; specifically, the exclusion of milk-based drinks and specification around eligibility of drinks mixed with carbon dioxide, water and ice, and those with and without milk.

> …they were looking at the likes of drinks fountains for carbonated soft drinks because… a bag and box syrup, they would be mixed with ice or carbon dioxide to give the carbonation or either they could be mixed with water and that would capture those drinks in the out-of-home sector, but there was a vagueness to milk-based drinks.—Out-of-home food and drink manufacturer

Some queries to Her Majesty's Revenue and Customs (HMRC—the tax collecting authority in the UK) went unanswered, thus, the out-of-home sector had to interpret the legislation themselves and apply the SDIL according to their interpretation. Representatives of the out-of-home sector did not perceive 2 years as enough time to have prepared due to confusion surrounding eligibility. In contrast, soft drinks manufacturers stated they had had time to prepare.

#### Challenges for supermarkets with large product portfolios
Supermarkets felt disadvantaged compared with soft drink manufacturers by the complexities of their sector. They highlighted sector-specific challenges to adapting to the SDIL, including that their product portfolio not only contains branded drinks, about which they have to make decisions, but also private label (own brand) drinks.

…what branded suppliers chose to do was their choice…different brands choosing to reformulate, resize or inflate, which I think led to a fair bit of customer confusion as to what the hell was going on.—Supermarket

It was described as challenging and time consuming to manage such a large portfolio and make decisions on each product. Particularly as reformulation decisions and portion size reduction reportedly differed between brands yet had to be merchandised together within stores. Retailers also felt that they were disadvantaged as their customers expressed confusion at differing responses by different brands—for example, 'sugary' drinks reformulated to just below the SDIL threshold but containing both sugar and sweeteners confused customers, with queries directed at retailers rather than drinks manufacturers.

… we tried to make it as clear for customers by putting on all the (shelves) sugar levy applied, so they could very much see…But…when they see a sugar line that's not (included in the SDIL), that's when the questions start coming.—Supermarket

### Theme 2: complex to implement but no lasting negative effects
#### Complexities in strategic response—price and product are key
Industry responded to the SDIL by reviewing product portfolios and strategically selecting responses at the individual product level. This portfolio review approach is why responses differed between companies and between products. Research and development (R&D) and consumer testing were costly for industry during this process, and, linking to theme 1, there were increased costs for those companies with larger product portfolios (eg, supermarkets). For the out-of-home sector, additional complications were noted due to confusion over the eligibility of some milk-based drinks.

…government is very keen to always say 'oh just reformulate, it will be easy' but it's not easy. It actually takes a lot of time and investment.—Drink manufacturer

Consumer testing was vital during the reformulation and decision-making process and consumer preference dictated the strategy taken.

… we invested a significant amount of money…in developing lots and lots of different formulations with lower sugar to see and testing them with consumers in Great Britain to see whether those recipes…would be acceptable to consumers.—Drink manufacturer

An additional challenge in reformulating drinks described by manufacturers was that sugar serves a functional purpose, in the mouthfeel of drinks mixed with ice and to prevent 'brain freeze', as well as to provide sweetness.

Because, actually, yes, we could stick sweeteners in everything, but, actually, sugar also has like a functional role.—Out-of-home retailer

Packaging, merchandising and placement were challenges to overcome, particularly for supermarkets. Decisions were made on own brand products but also on how to retail other branded products with different responses to the SDIL (eg, reformulated drinks, reduced and increased portion sizes, rebranding).

…there were a number of products that didn't reformulate but did drop size. So, again, there's just small considerations in that around how you merchandise it… So what sounds like a relatively simple change, of dropping from 330ml to, I don't know, 250ml, in reality kind of that complexity flows back through the value chain.—Supermarket

#### Leadership buy-in dictates strategic response
Leadership buy-in to health, where senior management 'buy-in' to the idea that their company should be making pro-health decisions, was discussed as vital in dictating the strategic response to the SDIL.

… I think such a review requires strong leadership and … our COO was very clear that we needed to step in and we needed to do, you know, do the responsible, brave thing.—Drink manufacturer

Participants described this buy-in as making the process simpler and a lack of buy-in as a barrier to making timely progress.

… having that strong leadership and, you know, complete buy-in from the top team and actually pretty much all the other levels of the organisation, then it's actually quite simple.—Drink manufacturer

#### Global companies and internal systems
The cost of setting up internal systems to account for and pay the SDIL was expensive, due to the requirement to report to HMRC, regardless of whether or not a company involved in the manufacture or selling of soft drinks was liable to pay the levy.

…It's ridiculous that, you know, it's cost us half a million pounds just to tell Treasury that actually we don't need to pay it.—Drink manufacturer

The global nature of many of these companies was an additional challenge. Response strategies appropriate for a UK market may not be transferable to other countries, for example, reformulation recipes vary due to differences in consumer palate and storage temperatures/facilities.

…that's (computer system) for the UK, and then Ireland have a separate system, France have a separate system, Mexico have a separate system.—Food and drink manufacturer

## Contradictory government messaging

There was confusion over whether manufacturers needed to pass on price increases to change consumer behaviour due to contradictory government messaging over the aim of the SDIL. Participants indicated that they thought price increases should have been passed on to target individual behaviour change; however, manufacturers stated they had no control over whether this occurred as retailers set the price for consumers

> …(the) government had slightly mixed messages so it was pretty clear from the Department of Health and PHE (Public Health England) … that they expected to see prices passed on … I think the Treasury were trying to say, oh soft drinks manufacturers don't have to pass this on… Well, apart from the fact that most businesses won't absorb a cost if they can avoid it for obvious reasons, it was the opposite of what the Health Department and others wanted…—Drink manufacturer

## Few long-lasting negative effects and the SDIL provided opportunities

Participants acknowledged that the SDIL did achieve its aim in stimulating product reformulation to avoid the levy. Although implementation was complex and costly, as previously illustrated, there were few long-lasting negative effects. Some participants suggested the SDIL provided opportunities.

> I think some of them would have switched back but we've gained new consumers as well which is, you know, how we, which through sampling and advertising essentially.—Drink manufacturer

However, participants were sceptical that the SDIL would achieve intended reductions in childhood obesity in the UK.

> … why (the SDIL) it was thought that that would be a, that policy in isolation would be sufficient to reduce obesity rates.—Drink manufacturer

## Theme 3: why us?—the SDIL unfairly targets the drinks industry

### Sugary drinks in isolation were unfair targets for regulation

Participants felt that the SDIL unfairly targets the soft drinks industry. Participants expressed their frustration that a single food category was targeted when other food categories bear a significant proportion of the responsibility for childhood obesity. They expressed the view that multiple nutrients or calories across many food and drink sectors should be targeted by regulation if the government is serious about reducing childhood obesity, particularly as substitution to other non-regulated food categories could negate the impact of the SDIL on health.

> …why would it be just the soft drink levy, why would you not target cakes and biscuits…that's what we

didn't understand at the time.—Food and drink manufacturer

There was consensus among participants that it did not make sense for the government to target a category that they considered was already reducing sugar faster than other food categories. Although the SDIL had accelerated the reformulation progress for some, this was stated to be already occurring prior to the SDIL announcement. Participants expressed the view that the sector had been unfairly penalised, and that sectors which reformulate should be praised rather than targeted by regulation when other unregulated categories have contributed little towards achieving health goals.

> … the soft drinks category was already well embarked on the journey to reformulation…part of the industry's disappointment and frustration about the announcement of the levy was that they were already absolutely going to deliver what the levy has now kind of made them deliver.—Trade association

### Distrust of government's motivations to introduce the SDIL

Participants stated that the SDIL was politically motivated, not an evidence-based policy. Government policies targeting obesity were described as contradictory and not aligned with one another, particularly the proposed ban on advertising of less healthy foods on TV and online.[34] According to participants, the advertising ban does not distinguish between reformulated and non-reformulated products, and acts as a disincentive to spending on reformulation if they cannot recoup their investment through advertising new products.

> So if you can take something from 40 g of sugar to 20 g of sugar but you'd only advertise on TV is it's 5 (g), then why bother, right, and it also means that they can't tell the world, look at this amazing thing we've done, we've reformulated this.—Advertising consultant

Perceived disconnectedness between policies led to distrust in the government and a belief that government obesity policy is poorly planned. Distrust was compounded by some companies appearing to be successful at lobbying the government following the announcement, resulting in changes to the regulations as a result of this lobbying, rather than on the basis of health or nutrition, in particular, the decision to exclude milk-based drinks. Participants stated this was motivated by some companies being able to gain a competitive advantage, as some milk-based drinks have higher sugar content than soft drinks. Participants also referred to the SDIL as a political tool to distract from other things in the budget in which it was announced.

> … I think this was a decision taken within the Treasury by quite a small group of people and it was announced during a Budget by a Chancellor who was trying to distract from some other economic figures

that he maybe wasn't too pleased about.—Drink manufacturer

The fact that the proposal to establish the SDIL had been kept secret, and the announcement was a shock to many, led to this view.

I think the timing was a surprise…Yeah and the way it was done without any form of consultation or pre-announcement.—Drink manufacturer

### Theme 4: the consumer is king
#### Consumer response to product changes resulting from the SDIL
Industry participants discussed throughout all previous themes that meeting the wishes of consumers was the priority when responding to the SDIL. Taste preferences and tolerance of reformulation changes were critical and companies expressed concerns that consumers might dislike reformulated products if they changed dramatically in a short time period.

… obviously what's critical from our perspective is developing a product that consumers still like the taste of while reducing their sugar intake so that we were trying to marry-up those two things.—Drink manufacturer

Company responses to the SDIL, as well as health and environmental issues more broadly, were vital to maintaining brand loyalty and company reputation in the eyes of consumers. The media were seen as influential in shaping consumer preferences and company reputation, as some newspapers had used graphics to show the sugar content of drinks and this was considered to have influenced purchasing patterns. According to informants, a small group of very loyal consumers can cause a backlash publicly, which can be picked up by both the news media and social media.

#### Consumer momentum towards healthier products
Participants stated consumer purchasing patterns are changing, with consumers increasingly choosing lower sugar products, which may also have driven reformulation prior to the SDIL. The policy acted as a catalyst for increasing consumer demand for sugar reduction and some respondents also highlighted the role of social media in driving these trends. Consumers were also reported by participants as 'moving away from' artificial ingredients, which led to challenges in reformulation using non-nutritive sweeteners

A lot of our consumers like…, they don't want to have sweeteners, they don't want to have preservatives.—Drink manufacturer

Some participants suggested that consumers were not lost when sugar was reduced in their favourite products, due to consumer preferences moving towards prioritising health. It was important to participants and their organisations that consumers have enough choice and there were concerns that regulation could limit choice from some.

### Theme 5: the future of the SDIL
Participants discussed the potential of expanding the SDIL to fruit and milk-based drinks, the wider threat to other products, reformulation in other categories, changes in other sectors as a result of the SDIL and the possibility of its reversal by government.

#### Extending to milk-based and fruit-based drinks
Concerns were expressed over the Chancellor's proposal to extend the SDIL to milk-based and fruit-based drinks at the time of the announcement.

I don't think politicians think it's done. Obviously we've got the review next year on whether milk-based drinks should be included, and then I think it's 2021 when they'll review the levels as well.—Drink manufacturer

Participants stated the nutritional benefits of these meant that natural sugars (fructose and lactose) should not be subject to the same regulation as soft drinks. The vitamin and mineral content of these drinks was also discussed as a benefit to children who may not be consuming sufficient fruit, vegetables or calcium from other sources.

… Now you have products that are being developed with high levels of sugar in them so that really does need to be addressed but you don't want to go down the route of demonising milk because it is still a great source of nutrition.—Out-of-home food and drink manufacturer

Reformulation of these drinks was considered particularly challenging, as naturally occurring sugars cannot be removed in the same way as added sugars in soft drinks.

#### Impact on the wider food and drink industry and on other sectors
A wider threat to other products, particularly those included in the PHE Sugar Reduction Strategy[35] (another element of the Childhood Obesity Plan that encouraged voluntary industry reformulation) was discussed. The SDIL demonstrated that the government was willing to implement policy to regulate the food industry in a way that has not been done before. Food and drink companies discussed their companies' attempts to reformulate products not included in the SDIL. The SDIL was described as a rallying call for industry to improve the healthfulness of products. It was also perceived to cause a ripple effect not just regarding health but also sustainability, environment, media and promotions.

Yeah, I think there is a ripple effect. So, I think it can be both positive and negative. I think in terms of positive, I think it can force companies to reformulate and be more innovative in driving the use of other ways of sweetening products—Food and drink manufacturer

## Proposal to reverse the SDIL

Comments made by Boris Johnson in his leadership campaign to become prime minister (July 2019) suggested he might consider repealing the SDIL.[36] These were not taken well by some participants; who indicated that companies had invested heavily in implementing the levy.

> I suppose it does feel like a backtrack (reversing the SDIL). Like we've made all this work and it was at the time quite painful in the sense of it was such a massive change through the supply chain so there was so many things to think about.—Out-of-home food and drink manufacturer

However, some participants suggested that reversing the SDIL would be well tolerated.

> I think, yeah, the industry would be happy to see the back of it because it's just cumbersome, it's just something, it's just another thing to administer.—Food and drink manufacturer.

# DISCUSSION
## Summary

Senior industry professional perspectives on the SDIL are described in five main themes. *Theme 1: a level playing field…for some, Theme 2: complex to implement, but no lasting negative effects, Theme 3: why us?—the SDIL unfairly targets the drinks industry, Theme 4: the consumer is king and Theme 5: the future of the SDIL.* The SDIL appeared to create a level playing field which industry accepted, however, this was perceived as inadequate due to the exclusion of milk-based drinks and targeting only SSBs, giving some a competitive advantage. Implementation of the SDIL was time consuming and complex, leading to high financial investment to prepare for it. Strategic response to the SDIL was dependent on leadership buy-in and particularly governed by potential consumer responses to product changes associated with the policy. The announcement and subsequent implementation of the SDIL caused a ripple effect beyond the soft drinks industry. The wider food and drink industry perceived it as evidence of the government being willing to regulate to help achieve health goals.

## Strengths and limitations

The use of elite interviewing techniques to build relationships with and solicit meaningful responses from participants is a strength of this work. These techniques allowed us to obtain the views of senior professionals from commercial organisations who have often been difficult to recruit to other studies.[37] As evident from the challenges described in the out-of-home sector and supermarkets, including respondents outside of manufacturing allowed wider exploration of the systemic impacts of the SDIL. A limitation of this work, however, is that interviews were carried out over a long period of time due to challenges in recruitment. Therefore, not all participants experienced the same political context, such as Boris Johnson's threats to reverse the SDIL in July 2019. Initial plans were for longitudinal data collection repeated across the time period of the study. Had all participants been interviewed closer to the implementation of the SDIL in 2018, then repeated in 2020, perspectives on the political events occurring would have been captured from all participants. Unfortunately, challenges to recruitment and access to elite participants led to the abandonment of this plan. Although researcher neutrality was expressed to participants the position of interviewers as public health academics could have led to these recruitment challenges.

The positionality of the researchers may also have led to censoring of responses by some participants. While we sought to descriptively represent industry perspectives, as well as acknowledge our own biases that are typically pro-health policy, it is important to acknowledge that the food and drink industry will have their own biases against health policy that is detrimental to their business survival, as evidenced in previous work.[38 39] Although it was not the aim of the work to explore participant responses in relation to the commercial determinants of health, it is possible that participant responses did not represent the reality of what occurred behind the scenes in the food and drink industry in relation to the SDIL. Overlap between some of the responses provided in this work and the 'typical' responses explored by other researchers as an industry 'playbook'[40] may support this assertion.

## Interpretation and implications

Interviewees reported that the technical aspects of drink production, particularly in the out-of-home sector, were not adequately accounted for in the design of the SDIL. An unintended consequence of the milk-based drink exclusion, led to some organisations having to interpret the particulars of the SDIL while their queries to HMRC went unanswered. Experiences of participants in this work align with findings that UK Government policy is set up poorly for the purposes of adequate monitoring and evaluation.[41] Future policy should engage with the wider food and beverage sector once a policy is certain to be implemented, to design and communicate technicalities in ways that avoid industry having to interpret themselves what is required and provide timely responses to queries surrounding implementation. Further, respondents indicated that lobbying against the inclusion of milk-based SSBs in the SDIL resulted in this exclusion. Alongside policy engagement in the technicalities of production, an avenue for future research would be to understand in more detail the policy process surrounding the SDIL, particularly the influence of the food and drink industry on the policy particulars.

Reviewing their product portfolio was also discussed, where assessments of the product mix as a whole and by individual product were conducted when determining the response to the SDIL. This aligns with previous findings that soft drink companies monitor their internal and

external contexts to determine their products' market position in response to a stimulus such as the SDIL, and then respond with marketing or non-marketing activity to influence the purchasing of soft drinks.[42] A crucial external contextual component to response in our findings appears to be consumer response and preferences towards each product, as well as health as a whole.

The UK soft drinks industry was reformulating products to lower sugar alternatives several years before the SDIL was introduced.[42 43] Perspectives expressed by participants align with this and suggest that there is a shift towards healthier drinks as the primary offer for consumers, with the SDIL accelerating the pace of this change. Consumer preferences for healthier products, and our finding that industry prioritises these health preferences in their decision making, are likely to have triggered the soft drinks industry to reformulate products prior to the announcement of the SDIL. The advocacy (eg, Jamie Oliver and Action on Sugar) in the early 2010s[44–46] and government threats to regulate industry[47] may have also increased consumer awareness about the health impact of sugar consumption and had a 'signalling effect' to consumers to reduce their sugar consumption.[48] Participants in our study suggested that the SDIL was adopted by the government because of the existing popularity of sugar reduction among the public. It is likely that the UK public was aware that SSBs harm health much earlier than the policy announcement, resulting from media activity, such as that related to Jamie Oliver's campaigning[44] PHE's[45] and WHO's reports on sugar.[46] Therefore, the importance of public momentum towards health could be regarded as a trigger for industry action independently from encouraging government action via policy.

Finally, participants expressed concern that policies introduced to combat obesity and other societal issues should be complementary not contradictory. The proposed ban on TV and online advertising of high fat, salt and sugar (HFSS) products by the UK Government[34] was viewed by industry to be misguided as they stated it may stop them from being able to advertise their reformulated products; not just those impacted by the SDIL but products voluntarily reformulated which would still be classified as HFSS. Stakeholder requests for consistency across policy areas were also expressed by interviewees regarding this advertising ban.[49] This indicates that a more consistent approach to determining which products government wants industry to change would help ensure policies do not undermine one another and build trust in government among industry.

## Conclusion

This study explored food and drink industry perspectives on the SDIL. We found that industry accepted that legislation was useful in levelling the commercially competitive playing field. However, in practice, participants stated that the SDIL had not created a 'true' level playing field as little consideration had been given to excluded product categories during policy design. Technical aspects of implementation were not adequately included and led to complexity for out-of-home retailers. Legislation on SSBs needs to take account of all industry sectors it affects, including out-of-home retail, as well as the manufacturing sector. Participants stated that only targeting sugary soft drinks was unfair due to the progress already made in the category compared with others (eg, confectionary). The critical role of consumers in creating momentum towards sugar reduction in SSBs prior to the SDIL announcement, as well as dictating response to the SDIL was discussed. It is hypothesised that pro-health public views could be a useful lever in encouraging positive industry action independently of food and drink regulation. The impact of the SDIL was felt beyond the soft drinks industry, driving other product sectors to reformulate in anticipation of future regulation.

**Author affiliations**
[1]MRC Epidemiology Unit, University of Cambridge, Cambridge, UK
[2]Nuffield Department of Primary Care Health Sciences, University of Oxford, Oxford, UK
[3]Global Food System and Policy Research, School of Kinesiology and Health Science, Faculty of Health, York University, Toronto, Ontario, Canada
[4]Population Health Innovation Lab, Department of Public Health, Environments & Society, Faculty of Public Health and Policy, London School of Hygiene & Tropical Medicine, London, UK
[5]Department of Agri-Food Economics and Marketing, University of Reading, Reading, UK
[6]Department of Social and Policy Sciences, University of Bath, Bath, UK
[7]Faculty of Health and Life Sciences, University of Exeter, Exeter, UK

**Acknowledgements** For the purpose of Open Access, the author has applied a Creative Commons Attribution (CC BY) licence to any Author Accepted Manuscript version arising.

**Contributors** CPJ: Conceptualization, Methodology, Formal analysis, Investigation, Data Curation, Writing - Original Draft, Project administration. HF: Formal analysis, Writing - Review and Editing. TLP: Conceptualization, Methodology, Investigation, Data Curation, Writing - Review and Editing, Funding acquisition. DT: Formal analysis, Writing - Review and Editing. SC: Conceptualization, Methodology, Formal analysis, Writing - Review and Editing, Supervision, Funding acquisition. JA: Formal analysis, Writing - Review and Editing, Funding acquisition. CL: Formal analysis, Writing - Review and Editing. HR: Formal analysis, Writing - Review and Editing, Funding acquisition. RS: Formal analysis, Writing - Review and Editing, Funding acquisition. MW: Conceptualization, Methodology, Resources, Supervision, Project administration, Funding acquisition, Formal analysis, Writing - Review and Editing, Guarantor.

**Funding** This project was funded by the NIHR Public Health Research programme (Grant Nos. 16/49/01 and 16/130/01). At the time this study was conducted CPJ, MW, ELR, HF, TLP, DT, JA, OA, SA, were also supported in part by: Programme grants to the MRC Epidemiology Unit from the Medical Research Council (grant No. MC_UU_12015/6 and MC_UU_00006/7); and the Centre for Diet and Activity Research (CEDAR), a UKCRC Public Health Research Centre of Excellence – funding from the British Heart Foundation, Cancer Research UK, the Economic and Social Research Council, the Medical Research Council, the National Institute for Health Research, and the Wellcome Trust, under the auspices of the UK Clinical Research Collaboration is gratefully acknowledged. HF received funding for her PhD studentship from the Economic and Social Research Council and Public Health England, and she has received further discretionary funding from the Economic and Social Research Council and Murray Edwards College, Cambridge. The views expressed are those of the authors and not necessarily those of the any of the above-named funders. The funders had no role in study design, data collection and analysis, decision to publish, or preparation of the manuscript.

**Competing interests** None declared.

**Patient and public involvement** Patients and/or the public were involved in the design, or conduct, or reporting, or dissemination plans of this research. Refer to the Methods section for further details.

**Patient consent for publication** Not applicable.

**Ethics approval** This study involves human participants. The study was approved by the Humanities and Social Science Ethics Committee at the University of Cambridge, UK. No approval ID was provided by this committee. Participants gave informed consent to participate in the study before taking part.

**Provenance and peer review** Not commissioned; externally peer reviewed.

**Data availability statement** No data are available. To maintain participant anonymity and confidentiality, data are not available due to the identifiable nature of participants from transcripts and recordings.

**ORCID iDs**
Catrin P Jones http://orcid.org/0000-0003-1425-0513
Steven Cummins http://orcid.org/0000-0002-3957-4357
Jean Adams http://orcid.org/0000-0002-5733-7830
Harry Rutter http://orcid.org/0000-0002-9322-0656

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
