## [Reviewer comments · BMJ Open]

This paper was submitted to a another journal from BMJ but declined for publication following peer review. The authors addressed the reviewers' comments and submitted the revised paper to BMJ Open. The paper was subsequently accepted for publication at BMJ Open.

ARTICLE DETAILS

TITLE (PROVISIONAL)	Industry views of the UK Soft Drinks Industry Levy: a thematic analysis of elite interviews with food and drink industry professionals, 2018-20
AUTHORS	Jones, Catrin; Forde, Hannah; Penney, Tarra; Theis, Dolly; Cummins, Steven; Adams, J; Law, Cherry; Rutter, Harry; Smith, Richard; White, Martin

VERSION 1 – REVIEW

REVIEWER	Elliott, Lana M. James Cook University, College of Public Health, Medical and Veterinary Science
REVIEW RETURNED	29-Apr-2023

GENERAL COMMENTS	The authors should be congratulated on this interesting body of research. Conducting research with industry is no easy feat, and the study's methods and findings demonstrate that the authors worked hard to foster trusting relationships and consciously analysed what was said with integrity. I have included point-by-point feedback below to support the authorship team in strengthening this paper and really drawing out its important contributions to the field: Strengths & Limitations: - Can the authors add another clause explaining the likely impact of the protracted recruitment/changing political context on their results? Introduction: - Line 24: 'was designed to incentivise manufacturers of SSBs to reformulate their products' – was this the only reason the tax was introduced or was it also introduced to shift consumption, accrue revenue?- The authors appear to have used two different referencing styles in this opening section. Can this be rectified?- Box 1: Can the authors provide some clarification on what is meant by 'alcohol replacement drinks'? Further, is the statement 'Manufacturers selling under one million litres of drinks per year' also linked to exemptions? If so, can this be reformatted to better reflect this. If this is the case, it'd also be great to see what
--

proportion of the market fit under this exemption and have this particular exemption was determined.

- Line 40/42: commencing 'This shift was...' – the phrasing 'purchases of sugar from SSBs' is a little hard to follow. It may be worth re-wording this sentence to improve clarity

Methods:

- Reflexivity subsection: Line 30/31 beginning 'the complete elimination...' I would suggest changing 'conducted' to 'achieved'

- Line 24: It may be worth considering alternative wording for 'put aside our biases and negativity...' Was it more that a process of reflexivity allowed the authors to acknowledge one's biases and account for them to some degree in how the research was conducted and analysed?

- Line 33: How was Braun and Clarke's thematic analysis approach modified? In explaining this, it may be worth separating out the clauses currently separated by a ; into two sentences.

- Line 50/51: 'May lean towards the critical' – can the authors please clarify this a little?

- Participants subsection: The principles of elite interviewing seem pretty consistent with how any well considered interviews should be conducted. Are there further points that separate this approach from others? I'm just conscious that this has been flagged as a standout approach of this study in the abstract but how it reads in this section doesn't quite reflect how it stands out from other potential approaches.

- Line 50/51 – 'c) they could provide a novel perspective...' Some further detail is potentially needed here to explain how the research team judged this. Was it about not doubling up on people from the same organisation/managerial level? It is worth explaining how the research team were able to assess whether their perspectives would or wouldn't be novel prior to interview.

- Analysis subsection: This subsection is well detailed, and it is easy to comprehend the steps involved and why the research team undertook them, well done.

- Patient and public involvement statement: Can the final sentence be adjusted to indicate what is meant by "project oversight"? Were the public involved in designing the study or just in providing guidance on the approach etc?

Results:

- Some results sub-section seems to provide an account followed by a series of quotes. I wonder if interspersing descriptions/analysis with quotes a little more may make for neater analyses.

- Theme 1: The SDIL created a level playing field subsection: Line 50/51 'The lack of consultation by government...' Can the authors provide clarification who participants meant by other sectors who were not soft drink manufacturers?

- Milk based drinks subsection: Line 44: It may be worth adjusting this first sentence to read 'Interviewees explained that, from their perspective, the government did not...'

- Line 37: "Thus, the out-of-home sector..." – Can this sentence be flipped around first indicating that queries went unanswered and subsequently that required that they interpret legislation themselves?

- Line 44: Representatives from the out-of-home section did not...

- Challenges for supermarkets subsection: Further detail is needed to explain how branded/private label drinks add additional challenges. This isn't clear from the detail provided about the SDIL in opening sections.

	 - Theme 2: Leadership buy-in subsection: Line 30 – Can the authors explain what is meant by ‘leadership buy-in to health’ or potentially consider rephrasing this sentence. - Contradictory government messaging subsection: Line 27 – Can this first sentence be amended, as it currently reads, passing on the tax appears in this sentence as almost an afterthought, but the next section indicates that it is quite central to the paragraph. - Theme 3: Why Us: Line 20: Sugar drink in isolation subsection: Line 20 – Please switch “didn’t” to “did not” in this sentence. - Line 30: “Existing direction of travel” sounds a little colloquial, can this be rephrased? - Distrust in government: A more detailed account of the policy process in the intro or early results (i.e. including what was enacted, what was adjusted early in implementation) would allow detail like milk exclusion’s ramifications on trust to be more obvious to the reader than trying to introduce what happened and explain its impact at the same time. - Line 30: “A small group of very loyal...”, it may be worth commencing this sentence with “According to informants...” or something similar. - Proposal to reverse the SDIL: References to Boris Johnson’s remarks are needed in the opening sentence of this section (Line 50). Discussion:  - The discussion section could be restructured at a number of points to more eloquently summarise findings, position them within the broader literature, and explain where to from here. The current subheading breakdown, at times, detracts from the really interesting findings your paper has found and how these gel with and sheds further light on the broader literature in this space. - This section again seems to employ two different referencing styles. Can this be rectified across the paper? - Relationship to prior knowledge: Please provide references to support advocacy mentioned on line 3-4 - Line 37: “Participants also discussed adopting...” This sentence is really long and would be worth splitting to ensure your argument is maintained. - Unanswered questions: This section could draw more heavily on supportive literature in explaining the gap/s that remain and why addressing them is of importance. Conclusion:  - This section seems to point-by-point rehash findings. This is of course great but integrating these themes more concisely and thinking about them in the context of the broader literature would be beneficial in this section and allow you to really highlight some points raised in your discussion.
--	--

REVIEWER	Huse, Oliver Deakin University Faculty of Health
REVIEW RETURNED	17-May-2023

GENERAL COMMENTS	Thank you for the opportunity to review this paper, entitled ‘Industry views of the UK Soft Drinks Industry Levy: a thematic analysis of elite interviews with food and drink industry professionals, 2018-20’. The paper is well written and methodically sound, and the authors should be commended. Overall, I suggest that the manuscript could be strengthened by tightening the methods and results sections, and being a little more critical in the
---

	discussion (though I understand that this is somewhat outside of the scope the paper). Background  - The background is thorough and provides a good overview of the SDIL. It would benefit from a brief description of SSB taxation policies – both their purpose(s) and the real-world evidence supporting their effectiveness. - The background may also benefit from a brief mention of industry’s unavoidable role in implementation and the subsequent importance of understanding their perspectives. Methods  - The authors have done a good job describing their qualitative methods in-depth. I do feel that there is some overlap between sections, and the methods section as a whole is quite long that could be shortened. For example, the focus on participant’s meaning over researcher interpretation is mentioned under ‘Methodological orientation’, ‘reflexivity’ and ‘analysis’. - Further, some sections (such as the reflexivity section) could be included in a supplementary file, as these are not integral to understanding the approach. Results  - The results section is clearly set out and supporting quotes are used well. However, it is also a long section. The authors might consider reducing the quantity of quotes included in text (just one would suffice in most cases, whereas two or even three are sometimes currently used). Additional quotes could be included in a table or supplementary file. - The authors could also reduce the length of some quotes through use of ellipses. Discussion  - I do not think that the relatively small sample size (14) of this research is a limitation as the purpose of qualitative research is not to collect large datasets but explore participant perceptions in-depth. However, I suggest that the authors consider mentioning this in the strengths and limitations section and perhaps highlighting an opportunity for future research. - I feel that a slightly more in-depth discussion of the health-harming nature of many of these corporations and the likely participant bias in responses would strengthen the paper. However, the authors make it clear that a discussion of the commercial determinants of health is outside the scope of this paper and this is understandable. I will leave it up to the authors as to whether there is room for a more critical discussion section. - I would suggest that the authors refer to the literature relating to industry reflexivity, bias, and a likely negative view of food and nutrition policies. This is mentioned, but I feel that more depth and support from the literature is required.
--	--

VERSION 1 – AUTHOR RESPONSE

Reviewer 1: Dr. Lana M. Elliott		
Abstract		

Introduction	Line 24: 'was designed to incentivise manufacturers of SSBs to reformulate their products' – was this the only reason the tax was introduced or was it also introduced to shift consumption, accrue revenue?	We have added a reference to George Osborne's budget speech which clarifies this he states the below: I "We're introducing the levy on the industry which means they can reduce the sugar content of their products – as many already do. It means they can promote low-sugar or no sugar brands – as many already are. They can take these perfectly reasonable steps to help with children's health. Of course, some may choose to pass the price onto consumers and that will be their decision, and this would have an impact on consumption too." Whilst he mentions some may pass on price of the levy and promote low sugar versions of drinks, the main motivation was to encourage reformulation. We have explicitly stated that this is how it was described in the budget speech.
	The authors appear to have used two different referencing styles in this opening section. Can this be rectified?	We have corrected this.
	Box 1: Can the authors provide some clarification on what is meant by 'alcohol replacement drinks'? Further, is the statement 'Manufacturers selling under one million litres of drinks per year' also linked to exemptions? If so, can this be reformatted to better reflect this. If this is the case, it'd also be great to see what proportion of the market fit under this exemption and have this particular exemption was determined.	We have amended 'replacement' to read 'substitute' drinks to make this clearer. The full regulations (https://www.legislation.gov.uk/ukxi/2018/41/made) would add significantly to the word count and we hope this change satisfies the reviewer. We have added a bullet point before 'manufacturers' which makes it clearer this is an exemption. Unfortunately, we do not have access to this information, the only detail on this and the exclusion can be found here - https://www.gov.uk/guidance/check-if-you-need-to-register-for-the-soft-drinks-industry-levy . We have added a citation to this website following the bullet point.
	Line 40/42: commencing 'This shift was...' – the phrasing 'purchases of sugar from SSBs' is a little hard to follow. It may be worth re-wording this sentence to improve	We have changed this sentence to start 'Reformulation is reflected in purchases of sugar...'

	clarity	
Methods	Reflexivity subsection: Line 30/31 beginning ‘the complete elimination...’ I would suggest changing ‘conducted’ to ‘achieved’	We have made this change and this section is now in supplementary file 1 as recommended by reviewer 2.
	Line 24: It may be worth considering alternative wording for ‘put aside our biases and negativity...’ Was it more that a process of reflexivity allowed the authors to acknowledge one’s biases and account for them to some degree in how the research was conducted and analysed?	We have amended this to “we sought to minimise the influence of our biases and negativity towards some of the practices of the food and drink industry, to truly ‘listen’ to the perspectives of our participants”. This section is now in supplementary file 1 as recommended by reviewer 2.
	Line 33: How was Braun and Clarke’s thematic analysis approach modified? In explaining this, it may be worth separating out the clauses currently separated by a ; into two sentences.	We have amended the sentence to read “As a result, a modified version of Braun and Clarke’s thematic analysis was used; reflexivity was a priority throughout the analysis in line with the approach however we sought to be less interpretive than their more recent guidance proposes [22].” This section is now in supplementary file 1 as recommended by reviewer 2.
	Line 50/51: ‘May lean towards the critical’ – can the authors please clarify this a little?	We have amended this to read “Whilst the researchers work to put aside their biases which may lean towards those more critical of the food and drink industry”
	Participants subsection: The principles of elite interviewing seem pretty consistent with how any well considered interviews should be conducted. Are there further points that separate this approach from others? I’m just conscious that this has been flagged as a standout approach of this study in the abstract but how it reads in this section doesn’t quite reflect how it	The steps may seem similar but ‘elite interviewing’ principles are more nuanced, e.g. there are lengthy discussions in the literature describing this method of the importance of understanding how to build rapport with very high-status individuals. Unfortunately, it is hard to distil the content of these works into concise enough steps to add to a methods section. We have though amended the sentence below in the hope this clarifies things somewhat. “The principles of elite interviewing were used to inform recruitment including stronger emphasis on the maintenance of trust, importance of interview tone of the interview, preparing appropriately, and engaging in and tailoring dialogue relevant to each informant, more so than in traditional interviews [25,26].”

	stands out from other potential approaches.	
	Line 50/51 – ‘c) they could provide a novel perspective...’ Some further detail is potentially needed here to explain how the research team judged this. Was it about not doubling up on people from the same organisation/managerial level? It is worth explaining how the research team were able to assess whether their perspectives would or wouldn’t be novel prior to interview.	We have amended this sentence to “c) they could provide a novel perspective, determined by their job role or the company they work for not previously heard in our interviews”
	Analysis subsection: This subsection is well detailed, and it is easy to comprehend the steps involved and why the research team undertook them, well done.	Thank you very much.
	Patient and public involvement statement: Can the final sentence be adjusted to indicate what is meant by “project oversight”? Were the public involved in designing the study or just in providing guidance on the approach etc?	We have amended this sentence to read “The ISSC for the overall project met biannually from 2017 – 2023 and were asked to provide advice on methodology as well as interpretation of our findings.”
Results	Some results sub-section seems to provide an account followed by a series of quotes. I wonder if interspersing descriptions/analysis with quotes a little more may make for neater analyses.	We have interspersed quotations within the text.
	Theme 1: The SDIL created a level playing field subsection: Line 50/51 ‘The lack of consultation by	We have amended this to read “The lack of consultation by the government with sectors who were not soft drinks manufacturers (for example out

	government...’ Can the authors provide clarification who participants meant by other sectors who were not soft drink manufacturers?	of home retailers) and the exclusion of milk-based sugary drinks led to this perception.”
	Milk based drinks subsection: Line 44: It may be worth adjusting this first sentence to read ‘Interviewees explained that, from their perspective, the government did not...’	We have made this amendment.
	Line 37: “Thus, the out-of-home sector...” – Can this sentence be flipped around first indicating that queries went unanswered and subsequently that required that they interpret legislation themselves?	We have changed this sentence to read “Some queries to Her Majesty’s Revenue and Customs (HMRC – the tax collecting authority in the UK) went unanswered, thus, the out-of-home sector had to interpret the legislation themselves and apply the SDIL according to their interpretation.”
	Line 44: Representatives from the out-of-home section did not...	We have made this amendment.
	Challenges for supermarkets subsection: Further detail is needed to explain how branded/private label drinks add additional challenges. This isn’t clear from the detail provided about the SDIL in opening sections.	We have added the following sentence to expand on this “Particularly as reformulation decisions and portion size reduction reportedly differed between brands yet had to be merchandised together within stores.”
	Theme 2: Leadership buy-in subsection: Line 30 – Can the authors explain what is meant by ‘leadership buy-in to health’ or potentially consider rephrasing this sentence.	We have included the following sentence to expand on this “where senior management ‘buy-in’ to the idea that their company should be making pro-health decisions”
	Contradictory government messaging subsection: Line 27 – Can this first sentence be amended, as it currently reads, passing on the tax	We have amended the sentence to read “There was confusion over whether manufacturers needed to pass on price increases to change consumer behaviour due to contradictory government messaging over the aim of the SDIL.”

	appears in this sentence as almost an afterthought, but the next section indicates that it is quite central to the paragraph.	
	Theme 3: Why Us: Line 20: Sugar drink in isolation subsection: Line 20 – Please switch “didn’t” to “did not” in this sentence.	We have made this amendment.
	Line 30: “Existing direction of travel” sounds a little colloquial, can this be rephrased?	We have amended this sentence to read “Although the SDIL had accelerated the reformulation progress for some, this was stated to be the already occurring prior to the SDIL announcement”
	Distrust in government: A more detailed account of the policy process in the intro or early results (i.e. including what was enacted, what was adjusted early in implementation) would allow detail like milk exclusion’s ramifications on trust to be more obvious to the reader than trying to introduce what happened and explain its impact at the same time.	We have included details of the public consultation process of the SDIL. Very little was adjusted during early implementation and the SDIL did not deviate from the plans set out in this consultation. Exclusions and the development of the policy appears to have been conducted behind closed doors.
	Line 30: “A small group of very loyal...”, it may be worth commencing this sentence with “According to informants...” or something similar.	We have made this amendment.
	Proposal to reverse the SDIL: References to Boris Johnson’s remarks are needed in the opening sentence of this section (Line 50).	We have made this amendment.
Discussion	Can the authors add another clause explaining the likely impact of the protracted	We have made substantial changes to the discussion integrating all the proposed changes where the original text referenced remains. We hope

	recruitment/changing political context on their results? The discussion section could be restructured at a number of points to more eloquently summarise findings, position them within the broader literature, and explain where to from here. The current subheading breakdown, at times, detracts from the really interesting findings your paper has found and how these gel with and sheds further light on the broader literature in this space. This section again seems to employ two different referencing styles. Can this be rectified across the paper? Relationship to prior knowledge: Please provide references to support advocacy mentioned on line 3-4 Line 37: "Participants also discussed adopting..." This sentence is really long and would be worth splitting to ensure your argument is maintained. Unanswered questions: This section could draw more heavily on supportive literature in explaining the gap/s that remain and why addressing them is of importance.	the reviewers agree that the section is now much improved.
Conclusions	This section seems to point-by-point rehash findings. This is of course great but integrating these themes more concisely and thinking	We have made substantial amendments to the conclusion to integrate this suggestion.

	about them in the context of the broader literature would be beneficial in this section and allow you to really highlight some points raised in your discussion.	
Reviewer 2: Dr. Oliver Huse		
Background	The background is thorough and provides a good overview of the SDIL. It would benefit from a brief description of SSB taxation policies – both their purpose(s) and the real-world evidence supporting their effectiveness.	We have made this amendment.
	The background may also benefit from a brief mention of industry's unavoidable role in implementation and the subsequent importance of understanding their perspectives.	We have made this amendment.
Methods	The authors have done a good job describing their qualitative methods in-depth. I do feel that there is some overlap between sections, and the methods section as a whole is quite long that could be shortened. For example, the focus on participant's meaning over researcher interpretation is mentioned under 'Methodological orientation', 'reflexivity' and 'analysis'.	We have moved the reflexivity section to a supplementary file which we believe improved this section.
	Further, some sections (such as the reflexivity section) could be included in a supplementary file, as these are not integral to understanding the approach.	We have included the reflexivity section as a supplementary file.
Results	The results section is clearly set out and supporting quotes are used well.	We have reduced the quantity and length of quotes as suggested.

	However, it is also a long section. The authors might consider reducing the quantity of quotes included in text (just one would suffice in most cases, whereas two or even three are sometimes currently used). Additional quotes could be included in a table or supplementary file.	
	The authors could also reduce the length of some quotes through use of ellipses.	
Discussion	I do not think that the relatively small sample size (14) of this research is a limitation as the purpose of qualitative research is not to collect large datasets but explore participant perceptions in-depth. However, I suggest that the authors consider mentioning this in the strengths and limitations section and perhaps highlighting an opportunity for future research.	We do not believe the sample size is a weakness of this study and adding this perpetuates the idea that generalisability, sample size and reliability are important components of qualitative research. Fundamentally these are incompatible with the qualitative paradigm. However, we have added that a longitudinal approach with repeated interviews with the same participants could be beneficial. Particularly to capture perspectives to the changing political landscape. We hope this extension to the limitations satisfies this element.
	I feel that a slightly more in-depth discussion of the health-harming nature of many of these corporations and the likely participant bias in responses would strengthen the paper. However, the authors make it clear that a discussion of the commercial determinants of health is outside the scope of this paper and this is understandable. I will leave it up to the authors as to whether there is room for a more critical discussion section.	Thank you for this helpful suggestion. As well as this paper we are reanalysing the data using commercial determinants of health theory and this lens in particular. We would like to keep this separate from the current paper and focus mostly on representing industry perspectives. We have however included more commentary on commercial determinants of health beyond what we had originally.

	I would suggest that the authors refer to the literature relating to industry reflexivity, bias, and a likely negative view of food and nutrition policies. This is mentioned, but I feel that more depth and support from the literature is required.	We have made substantial amendments to the discussion and have integrated this suggestion within these.
--	---	--

VERSION 2 – REVIEW

REVIEWER	Elliott, Lana M. James Cook University, College of Public Health, Medical and Veterinary Science
REVIEW RETURNED	15-Jun-2023
GENERAL COMMENTS	Congratulations to the authorship team on such a robust response to reviewer feedback and I hope you feel the process has strengthened the articulation of your research findings. I have no substantial suggestions for further revisions.